# Predictors of High-Burden Residual Axillary Disease After Neoadjuvant Therapy in Breast Cancer

**DOI:** 10.3390/cancers17101596

**Published:** 2025-05-08

**Authors:** Damiano Gentile, Jacopo Canzian, Erika Barbieri, Andrea Sagona, Simone Di Maria Grimaldi, Corrado Tinterri

**Affiliations:** 1Breast Unit, IRCCS Humanitas Research Hospital, Via Manzoni 56, Rozzano, 20089 Milan, Italy; erika.barbieri@cancercenter.humanitas.it (E.B.); andrea.sagona@cancercenter.humanitas.it (A.S.); simone.dimariagrimaldi@cancercenter.humanitas.it (S.D.M.G.); corrado.tinterri@hunimed.eu (C.T.); 2Department of Biomedical Sciences, Humanitas University, Via Rita Levi Montalcini 4, Pieve Emanuele, 20090 Milan, Italy; jacopo.canzian@humanitas.it; 3Medical Oncology and Hematology Unit, IRCCS Humanitas Research Hospital, Via Manzoni 56, Rozzano, 20089 Milan, Italy

**Keywords:** breast cancer, neoadjuvant therapy, axillary lymph node dissection, sentinel lymph node biopsy, axillary burden prediction

## Abstract

This study aimed to identify factors associated with high-burden residual axillary disease in breast cancer patients undergoing neoadjuvant therapy (NAT). Among 262 patients treated with NAT followed by axillary lymph node dissection (ALND), baseline cN+ status, HR+/HER2− tumor subtype, and larger post-NAT tumor size emerged as independent predictors of ypN2-3 disease. Despite being the standard approach when residual axillary disease is present, ALND has not demonstrated a significant survival benefit compared to less invasive strategies but is associated with increased complications. Achieving an axillary pathological complete response (pCR) remains a key prognostic factor for favorable long-term outcomes. Identifying patients at higher risk of residual disease may allow for optimization of preoperative systemic treatments to increase the probability of achieving axillary pCR, potentially sparing patients from ALND and its related morbidity while improving survival outcomes.

## 1. Introduction

Neoadjuvant therapy (NAT) is now an integral part of the treatment of breast cancer (BC), especially in locally advanced or high-risk disease [1,2,3]. Originally developed to enhance resectability in non-resectable tumors [4,5], NAT is currently extensively utilized to downstage the primary tumor and axillary lymph node involvement so that less extensive surgery can be performed and treatment response can be evaluated in vivo [6,7]. The development of targeted agents such as trastuzumab, pertuzumab, and immune checkpoint inhibitors has significantly increased response rates, especially in HER2+ and triple-negative BC subtypes [8,9]. The overarching goal is to minimize the extent of surgery in both the breast and the axilla, while ensuring oncologic safety. Downstaging of the tumor in the breast typically makes breast-conserving surgery (BCS) feasible instead of mastectomy [10,11,12]. Similarly, for axillary management, the traditional procedure of axillary lymph node dissection (ALND) is decreasing in favor of sentinel lymph node biopsy (SLNB). SLNB has long been the standard for clinically node-negative (cN0) patients, but increasing evidence supports its application in clinically node-positive (cN+) patients with a good radiologic and clinical response after NAT [13,14]. Large prospective trials, including ACOSOG Z1071 [15], SENTINA [16], and SN FNAC [17], have established that SLNB can be performed with acceptable false-negative rates in patients who convert from cN+ to cN0 after NAT. The omission of ALND in patients who become ypN0 after NAT is now considered safe, even in patients who initially had nodal involvement [18,19]. No increase in rates of recurrence or worse survival rates has been seen in patients who avoid ALND after achieving axillary pathologic complete response (pCR) [20,21,22]. However, despite this trend of de-escalation, ALND is still necessary in the presence of residual macrometastases (ypN+) in the sentinel lymph node (SLN) after NAT [23]. Persistent nodal disease following NAT represents a serious concern, being a marker for systemic treatment resistance and associated with increased recurrence rates and worse survival outcomes [24,25]. This presents a significant challenge, as patients with ypN+ disease after NAT often require intensified post-operative systemic and radiation therapy to improve outcomes. Interestingly, many studies have shown that more aggressive axillary surgery, such as ALND, has not provided fewer recurrence rates and improved survival but is instead paralleled by significant morbidity, including lymphedema, shoulder disability, sensory deficit, and chronic pain [25,26]. Given both the lack of clear oncologic benefit and the high risk of complications, there is an urgent need to better stratify patients who are truly at risk of harboring high-burden residual axillary disease after NAT. Identifying accurate predictive factors would help optimize patient selection for ALND and allow clinicians to tailor neoadjuvant strategies aimed at achieving nodal clearance. Both breast and axillary pCR are now the best prognostic indicators in support of having good long-term results in patients with BC treated with NAT [27]. A number of studies have established that pCR is associated with decreased recurrence rates and improved overall survival [28,29]. Nevertheless, there is limited real-world data specifically focusing on predictors of high-volume residual nodal disease (ypN2-3) in patients undergoing ALND after NAT. This knowledge gap hinders progress toward more personalized and effective treatment approaches. To address this unmet clinical need, we conducted a retrospective analysis of BC patients undergoing ALND after NAT. Our patients were stratified into two groups based on residual nodal disease burden: low-disease burden (ypN0, ypNmi, or ypN1) and high-disease burden (ypN2 or ypN3). The primary objective of our study was to identify predictors of high-disease burden (ypN2-3) following NAT, which could aid in better stratification of patients for more aggressive preoperative therapy.

## 2. Materials and Methods

### 2.1. Study Population

This retrospective study analyzed all the consecutive BC patients who underwent NAT followed by ALND at the Breast Unit of IRCCS Humanitas Research Hospital, Milan, Italy, from October 2006 to December 2023. Patients included in the study were diagnosed with invasive BC and underwent systemic therapy before surgery as part of their multidisciplinary treatment. Data were retrieved from an institutional database and electronic medical records, which included detailed demographic information, tumor characteristics, and treatment-related variables. The study included both cN+ and cN0 patients at the baseline, and all of them underwent ALND after NAT. Baseline tumor assessment included a comprehensive radiologic evaluation in the form of breast and axillary ultrasound (US) in all patients, mammography in most cases, and magnetic resonance imaging (MRI) or positron emission tomography (PET) when indicated. Histologic diagnosis was established with core needle biopsy of the primary breast tumor and, if necessary, fine-needle aspiration or core biopsy of suspicious axillary nodes. The tumors were classified based on biologic subtype, hormone receptor (HR), and HER2 status, with standard pathology criteria [30]. Patients received NAT with anthracycline- and taxane-based chemotherapy, with the addition of targeted therapies (trastuzumab, pertuzumab, or pembrolizumab) administered when indicated. Definitive treatment with NAT regimens was discussed in a multidisciplinary tumor board. Following NAT, all the patients subsequently underwent surgery consisting of either BCS or mastectomy for primary tumor management. Axillary staging and treatment were performed through ALND in all cases, either as a direct procedure or following SLNB. Postoperative treatment, including radiotherapy and adjuvant systemic therapy, was recorded.

### 2.2. Axillary Surgery and Pathological Assessment

Axillary surgery consisted of ALND in all patients, either as a direct procedure or following SLNB when performed. In patients undergoing SLNB, a single-tracer radiocolloid technique was used for lymphatic mapping. The number of sentinel and non-SLNs retrieved, as well as the total nodal burden, were recorded for all cases. Axillary nodal response was classified based on the post-NAT nodal stage (ypN), with patients stratified into low-disease burden (ypN0, ypNmi, or ypN1) and high-disease burden (ypN2 or ypN3). Pathological examination of lymph node specimens was performed by dedicated breast pathologists.

### 2.3. Statistical Analysis

Statistical analyses were conducted to evaluate factors associated with residual high-burden nodal disease (ypN2-3) post-NAT. Descriptive statistics were used to describe patient demographics, tumor characteristics, and treatment details. Categorical variables were reported as frequencies and percentages, while continuous variables were reported in the form of means and standard deviations (SD) or medians and ranges, as appropriate. Comparisons between low-disease burden (ypN0-mi-1) and high-disease burden (ypN2-3) groups were performed using chi-square tests for categorical variables and independent t-tests or Mann–Whitney U tests for continuous variables. Variables that were statistically significant in univariate analysis (*p* < 0.05) were subsequently included in a multivariate logistic regression model to identify independent predictors of residual high-burden nodal disease post-NAT. Odds ratios (OR) and corresponding 95% confidence intervals (95%CI) were calculated. To determine the predictive performance of significant variables, a receiver operating characteristic (ROC) curve analysis was performed. The area under the curve (AUC) was calculated to determine the ability of the predictive model to discriminate between patients with ypN0-mi-1 and ypN2-3 disease. A higher AUC value indicated stronger discriminatory power. Two-sided statistical tests were performed, and a *p*-value of <0.05 was considered statistically significant. Statistical analyses were conducted using IBM SPSS Statistics for Windows, Version 25.0. Armonk, NY, USA.

## 3. Results

### 3.1. Baseline Characteristics

A total of 262 BC patients who underwent NAT followed by ALND were analyzed. The mean age of the patients was 53.1 years (SD: 11.9), with 158 patients (60.3%) being post-menopausal. Preoperative imaging was performed in all cases, with 180 patients (68.7%) undergoing mammography, while breast and axillary US was performed in all patients. MRI was performed in 105 patients (40.1%), and 189 patients (72.1%) underwent PET. The mean tumor size before NAT was 36.3 mm (SD: 17.8). Regarding pre-NAT staging, 191 patients (72.9%) were diagnosed with cT1-2 disease, while 71 patients (27.1%) presented with cT3-4 tumors. The vast majority of patients (241; 92.0%) had cN+ disease, whereas only 21 patients (8.0%) were cN0. Most patients (250; 95.4%) received anthracycline- and taxane-based NAT. Anti-HER2 therapy with trastuzumab was administered in 84 patients (32.1%), while pertuzumab and pembrolizumab were given to 6 (2.3%) and 8 (3.0%) patients, respectively. The most common tumor subtype was HR+/HER2−, accounting for 45.8% of cases, followed by HER2+ disease (32.1%) and triple-negative BC (22.1%). The majority of tumors were ductal carcinomas (91.2%), and 72.5% of patients presented with a single tumor nodule. After NAT, the mean tumor size was 19.0 mm (SD: 25.5). Breast surgical treatment included BCS in 111 patients (42.4%) and mastectomy in 151 patients (57.6%). Regarding post-operative treatment, 223 patients (85.1%) received radiotherapy, 163 patients (62.2%) underwent endocrine therapy, and trastuzumab emtansine (T-DM1) was administered to 55 patients (21.0%). The full baseline characteristics of the study population are reported in Table 1.

### 3.2. Axillary Surgical Approaches and Nodal Staging Post-Neoadjuvant Therapy

Overall, 87 patients (33.2%) undergoing SLNB subsequently proceeded to ALND, while 175 patients (66.8%) had direct ALND. The median number of SLNs retrieved was 1 (range, 1–6). Among the 87 patients who underwent SLNB, 50 (57.5%) had a single SLN removed, 19 (21.8%) had two SLNs, and 18 (20.7%) had three or more SLNs. The median number of non-SLNs evaluated was 12 (range, 3–49), with a median of 1 metastatic non-SLN (range, 0–34). Following NAT, 66 patients (25.2%) achieved ypN0 status, while 10 (3.8%) were classified as ypNmi. Macrometastatic residual nodal disease was found in 186 patients (71.0%), with 92 patients (35.1%) classified as ypN1, 61 (23.3%) as ypN2, and 33 (12.6%) as ypN3. The axillary data and staging details are summarized in Table 2.

### 3.3. Predictors of Residual High-Burden Axillary Disease

To identify factors predictive of high residual axillary disease burden post-NAT, patients were divided into two groups: 168 patients (64.1%) had ypN0-mi-1 disease, whereas 94 patients (35.9%) had ypN2-3 disease. Univariate analysis revealed significant differences between the two groups. Patients treated with immunotherapy were significantly more likely to have low-burden residual axillary disease than those who were not (*p* < 0.001). Larger pre- and post-NAT tumor sizes were associated with higher residual nodal burden (mean 39.9 mm in the ypN2-3 group vs. 34.3 mm in the ypN0-1 group, *p* = 0.034, and mean 29.9 mm in the ypN2-3 group vs. 12.9 mm in the ypN0-1 group, *p* < 0.001, respectively). cT3-4 and cN+ patients were at a significantly higher risk of presenting with ypN2-3 disease than cT1-2 and cN0 patients (*p* = 0.029 and *p* = 0.031, respectively). Additionally, patients with HR+/HER2− subtype were more frequently found in the ypN2-3 group (*p* < 0.001). Conversely, HER2+ subtype was significantly higher in patients with ypN0-mi-1 disease (*p* < 0.001). The type of breast surgery and number of evaluated non-SLNs did not reach statistical significance. In multivariate analysis, cN+ at diagnosis remained the strongest independent predictor of residual high-burden nodal disease post-NAT, with an OR of 7.697 (95%CI: 1.537–38.550, *p* = 0.013). HR+/HER2− subtype was also independently associated with ypN2-3 status, with an OR of 3.945 (95%CI: 1.602–9.716, *p* = 0.003). Additionally, larger post-NAT tumor size was a significant risk factor, with an OR of 1.043 (95%CI: 1.021–1.066, *p* < 0.001). These findings indicate that baseline nodal involvement, HR+/HER2− subtype, and post-NAT tumor size are the most important predictors of residual high-burden axillary disease following NAT. Full details of the univariate and multivariate analyses are presented in Table 3.

### 3.4. Predictive Performance of the Model

To assess the predictive performance of these factors, a ROC curve analysis was performed. The AUC was 0.818, indicating strong discriminatory ability in distinguishing patients with ypN0-mi-1 from those with ypN2-3 disease. The ROC curve is displayed in Figure 1.

## 4. Discussion

### 4.1. Main Predictors of Residual High-Burden Disease and Comparative Evidence from Predictive Models and Nomograms

We identified baseline cN+ status, HR+/HER2− tumor subtype, and larger post-NAT tumor size as independent predictors of residual high-burden axillary disease after NAT. Other previous studies have also addressed this topic, with aims to identify predictors of axillary pCR or additional nodal tumor burden after NAT and to develop predictive models or nomograms to guide axillary management. Jin et al. [31] analyzed a cohort of 426 BC patients with biopsy-proven node-positive disease who received NAT, aiming to identify predictors of axillary pCR and to develop a predictive nomogram. In their cohort, the overall axillary pCR rate was 30.0% (128/426 patients). Multivariate logistic regression identified HR positivity as a strong negative predictor of axillary pCR (OR = 0.162, 95%CI: 0.074–0.353, *p* < 0.001), while HER2+ patients treated with trastuzumab had a significantly higher probability of achieving nodal pCR compared to HER2− patients (OR = 3.443, 95%CI: 1.178–10.060, *p* = 0.024). Tumor size also played a role, with smaller tumors (T1) associated with higher pCR rates compared to larger tumors (T3 OR = 0.195, 95%CI: 0.082–0.465, *p* < 0.001; T4 OR = 0.304, 95%CI: 0.138–0.670, *p* = 0.003). With these variables, the authors developed and validated a predictive nomogram with an AUC of 0.804 in the training cohort and 0.749 in the validation cohort and good discriminatory ability. Kantor et al. [32] conducted a large retrospective analysis using the National Cancer Database to investigate predictors of axillary pCR in patients with cN+ BC treated with NAT. Among 19,115 women registered from 2010 to 2013, 27.3% achieved axillary pCR and 17.6% achieved breast pCR. The authors developed and validated a predictive model based on pre-treatment clinical and pathological variables. In the multivariate analysis, several independent predictors of axillary pCR were identified: Younger age (<50 years) had higher odds of pCR (OR = 1.41, *p* < 0.05), and tumor subtype had a significant impact on the probability of nodal clearance. Specifically, HR-/HER2+ tumors had the highest probability of pCR (OR = 5.51, *p* < 0.05), followed by HR+/HER2+ (OR = 3.67, *p* < 0.05) and triple-negative (HR-negative/HER2−negative) tumors (OR = 2.80, *p* < 0.05). Corsi et al. [33] conducted a large multicenter retrospective analysis on 1950 cN+ BC patients treated with NAT between 2005 and 2020 across 11 Italian breast units. The primary objective was to identify independent predictors of axillary pCR and to develop a preoperative nomogram to select patients suitable for axillary de-escalation. Axillary pCR was observed in 886 patients (45.4%), while 1064 patients (54.6%) had residual nodal disease after surgery. The majority of the patients (85.8%) were originally cN1, while cN2 and cN3 stages were less frequent (10.6% and 3.6%, respectively). Multivariate logistic regression identified several independent predictors of axillary pCR. Axillary clinical complete response after NAT was the strongest predictor (OR = 2.95, 95%CI: 2.36–3.68, *p* < 0.0001). Tumor biology played a significant role as well: patients with ER-/HER2+ tumors had the greatest likelihood of nodal pCR (OR = 3.34, 95%CI: 2.02–5.52, *p* < 0.0001), followed by ER+/HER2+ tumors (OR = 2.40, 95%CI: 1.58–3.65, *p* < 0.0001) and triple-negative tumors (OR = 1.94, 95%CI: 1.34–2.81, *p* = 0.0004). Based on these factors, a nomogram was built and internally validated (AUC = 0.77, 95%CI: 0.75–0.80) with a sensitivity of 71% and specificity of 73%. Sanders et al. [34] conducted a single-institution retrospective analysis at Mayo Clinic with 229 BC patients treated with NAT who had positive SLN involvement and then underwent ALND. Of these, 177 patients (77.3%) were initially cN+ and 52 (22.7%) were cN0 prior to NAT. The primary endpoint was to assess the rate of non-SLN positivity and to identify predictors of additional nodal disease. Overall, 56.3% (129/229) of patients had additional positive non-SLNs at ALND. The rate was numerically higher among cN+ patients (59.3%) compared with cN0 patients (46.2%), but this was not found to be statistically significant (*p* = 0.09). In the cN0 subgroup, no clinicopathologic factor was significantly associated with additional non-SLN involvement. However, in cN+ patients, several factors were independently associated with residual axillary disease on multivariate logistic regression. HER2− status was associated with higher risk of positive non-SLNs compared to HER2+ tumors (OR = 2.52, *p* = 0.04). Multifocal or multicentric disease increased the risk of non-SLN positivity (OR = 2.08, *p* = 0.03).

### 4.2. Validation of the Predictive Role of HR+/HER2− Tumors

As demonstrated by our study and consistently corroborated by previous research, HR+/HER2− tumor subtype is a common and strong predictor of residual high-burden axillary disease after NAT. In our cohort, HR+/HER2− tumors were significantly associated with an increased risk of residual ypN2-3 disease, showing an adjusted OR of 3.945 (95%CI: 1.602–9.716, *p* = 0.003). The study by Schipper et al. [35] provides interesting data on the axillary response to neoadjuvant endocrine therapy (NET) in patients with HR+/HER2− BC and cN+ disease. Among 561 patients who were treated with NET between 2014 and 2019, the axillary pCR rate was extremely low, with only 7.3% (41/561) of patients achieving nodal clearance, and just 2.5% (14/561) experiencing concurrent breast and axillary pCR. Most patients had significant residual axillary disease after NET, with 67.6% classified as ypN1, 17.6% as ypN2, and 7.5% as ypN3. Friedman-Eldar et al. [36] investigated axillary response to NAT in 176 patients with HR+/HER2− cN+ BC treated between 2011 and 2020. The overall axillary pCR rate was low, at 12.3% (22/178 cases). Notably, younger patients and those receiving NET for more than 6 months were more likely to achieve axillary pCR, although the absolute rates remained low. Barbieri et al. [37] analyzed 205 patients with HR+/HER2− cN+ BC who underwent NAT between 2008 and 2019. Consistent with previous literature, HR+/HER2− tumors showed limited sensitivity to NAT, with a pCR rate of only 16.6% (34/205) in the breast and 21.5% (44/205) in the axilla.

### 4.3. The Role and Limitations of Axillary Lymph Node Dissection

Unfortunately, in case these risk factors are present, and patients present with residual axillary disease or a high nodal burden following NAT, completion ALND remains the standard of care. Despite its pivotal role in providing accurate staging, numerous studies have consistently shown that ALND does not result in a significant oncologic benefit in terms of disease-free or overall survival compared to less invasive techniques when there is minimal residual axillary disease. Dux et al. [26] conducted a prospective cohort study evaluating the oncologic outcomes of 292 patients with biopsy-proven cN+ BC treated with NAT, comparing patients who received ALND to those treated with targeted axillary surgery (TAS) for residual nodal disease (ypN+). Among the ypN+ subgroup, 75% underwent ALND and 25% received TAS. Despite the traditionally recommended use of ALND, the study found no significant oncologic advantage associated with its use. In fact, axillary recurrence was rare and did not significantly differ between groups: No axillary recurrences were observed in the TAS group, while four cases occurred in the ALND group (*p* = 0.21). Likewise, five-year axillary recurrence-free survival (100% vs. 90%, *p* = 0.21), overall survival (97% vs. 85%, *p* = 0.39), and disease-free survival (51% vs. 61%, *p* = 0.9) were comparable between TAS and ALND patients. Tinterri et al. [25] conducted a retrospective analysis of 322 cN+ BC patients who underwent NAT and converted to ycN0 status before surgery. Patients underwent either ALND or SLNB. With a median follow-up of 75 months, the study showed that SLNB not only resulted in comparable axillary recurrence rates to ALND (0.9–2.1%) but was also associated with significantly improved long-term oncological outcomes. Specifically, SLNB patients experienced higher 3-, 5-, and 10-year recurrence-free survival (*p* = 0.001), distant disease-free survival (*p* = 0.001), overall survival (*p* = 0.001), and BC-specific survival (*p* = 0.002). Multivariate analysis confirmed that ALND was independently associated with worse recurrence-free survival (HR = 0.356, *p* = 0.001), distant disease-free survival (HR = 0.376, *p* = 0.002), and BC-specific survival (HR = 0.228, *p* = 0.004), while ypN0 status and achieving a pCR remained favorable prognostic factors.

### 4.4. Study Limitations

This study is limited by its retrospective design and single-institution setting, which may limit the generalizability of the results. Although the data were collected from a prospectively maintained institutional database, inherent biases related to patient selection, treatment allocation, and data completeness cannot be excluded. Furthermore, the analysis did not systematically include relevant tumor biological markers such as the Ki67 proliferation index, which have been reported by previous studies as significant predictors of axillary response after NAT. The lack of integration of these biomarkers into the multivariate model may have limited the identification of additional prognostic factors capable of refining patient stratification.

## 5. Conclusions

In conclusion, we identified baseline cN+ status, HR+/HER2− tumor subtype, and larger post-NAT tumor size as independent predictors of high-burden residual axillary disease (ypN2-3) following NAT. Although ALND is still required in the presence of such residual disease, its ability to improve long-term oncologic outcomes remains unproven, as no significant survival benefit has been demonstrated compared to less invasive axillary approaches. On the other hand, ALND is invariably associated with an increased risk of surgical morbidity and functional impairment. Given the prognostic importance of achieving an axillary pCR, future efforts should focus on improving the identification of patients at high risk of residual axillary disease before surgery. Preoperative systemic therapies can be maximized through the application of intensified or targeted regimens, which may increase the probability of nodal clearance in these patients. This strategy potentially can reduce the need for ALND, minimize treatment-related morbidity, and improve oncologic results by treating more patients with the prognostic advantage observed with axillary pCR.

## Figures and Tables

**Figure 1 cancers-17-01596-f001:**
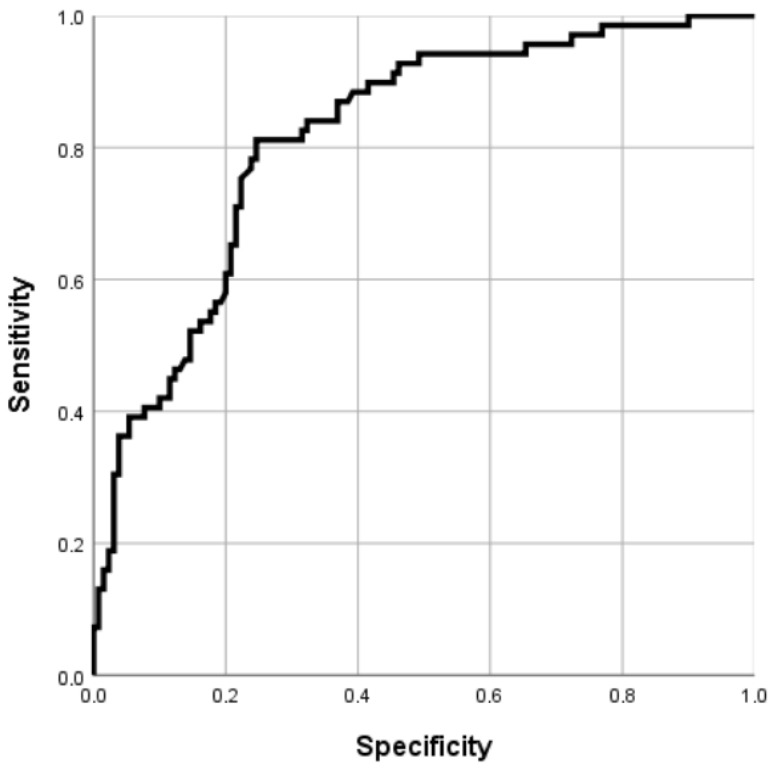
Receiver operating characteristic (ROC) curve for the prediction of residual high-burden axillary disease (ypN2-3) after neoadjuvant therapy. Footnotes: The ROC curve illustrates the discriminatory performance of the multivariate model in identifying patients with high-burden residual axillary disease (ypN2-3) following neoadjuvant therapy. The model included baseline cN+ status, HR+/HER2− tumor subtype, and post-NAT tumor size as independent predictors. The area under the curve (AUC) was 0.818, indicating good predictive accuracy.

**Table 1 cancers-17-01596-t001:** Baseline characteristics of 262 breast cancer patients undergoing neoadjuvant therapy and treated by axillary lymph node dissection.

Characteristics	Number (%)/Mean (SD)
**Demographics**	
Age (years)	53.1 (11.9)
Post-menopausal	158 (60.3%)
**Pre-operative staging**	
Mammography	180 (68.7%)
Breast and axillary US	262 (100%)
Axillary biopsy	102 (38.9%)
MRI	105 (40.1%)
PET	189 (72.1%)
Dimension pre-NAT (mm)	36.3 (17.8)
Stage pre-NAT	
cT1	39 (14.9%)
cT2	152 (58.0%)
cT3	36 (13.7%)
cT4	35 (13.4%)
cN0	21 (8.0%)
cN+	241 (92.0%)
**Neoadjuvant therapy**	
NAT without anthracycline	12 (4.6%)
NAT with anthracycline and taxanes	250 (95.4%)
Trastuzumab	84 (32.1%)
Pertuzumab	6 (2.3%)
Pembrolizumab	8 (3.0%)
Completed cycles	227 (86.6%)
**Tumor**	
Subtype	
HR+/HER2−	120 (45.8%)
HER2+	84 (32.1%)
Triple-negative	58 (22.1%)
Histotype	
Ductal	239 (91.2%)
Lobular	23 (8.8%)
Single nodule	190 (72.5%)
**Pathologic response**	
Dimension post-NAT (mm)	19.0 (25.5)
Stage post-NAT	
ypT0	54 (20.6%)
ypTis	13 (5.0%)
ypTmi	4 (1.5%)
ypT1a	18 (6.9%)
ypT1b	22 (8.4%)
ypT1c	60 (22.9%)
ypT2	65 (24.8%)
ypT3	4 (1.5%)
ypT4	22 (8.4%)
**Surgical treatment**	
BCS	111 (42.4%)
Mastectomy	151 (57.6%)
**Post-operative treatment**	
Taxanes	21 (8.0%)
Capecitabine	28 (10.7%)
Radiotherapy	223 (85.1%)
Endocrine	163 (62.2%)
T-DM1	55 (21.0%)
Abemaciclib	9 (3.4%)

Footnotes: SD: Standard deviation, US: Ultrasound, MRI: Magnetic resonance imaging, PET: Positron emission tomography, NAT: Neoadjuvant therapy, HR: Hormone receptor, HER2: HER2 evaluated either on immunohistochemistry or on in situ hybridization, according to the ASCO CAP guidelines, BCS: Breast-conserving surgery, T-DM1: Trastuzumab emtansine.

**Table 2 cancers-17-01596-t002:** Axillary data and staging in 262 breast cancer patients undergoing neoadjuvant therapy and treated by axillary lymph node dissection.

	Number (%)/Median (Range)
**Type of axillary surgery**	
SLNB followed by ALND	87 (33.2%)
Direct ALND	175 (66.8%)
**Number of SLNs**	1 (1–6)
1	50/87 (57.5%)
2	19/87 (21.8%)
≥3	18/87 (20.7%)
**Data on non-SLNs**	
Number of evaluated non-SLNs	12 (3–49)
Number of metastatic non-SLNs	1 (0–34)
**Nodal stage post-NAT**	
ypN0	66 (25.2%)
ypNmi	10 (3.8%)
ypN1	92 (35.1%)
ypN2	61 (23.3%)
ypN3	33 (12.6%)

Footnotes: SLNB: Sentinel lymph node biopsy, ALND: Axillary lymph node dissection, SLN: Sentinel lymph node.

**Table 3 cancers-17-01596-t003:** Predictors of residual nodal disease in 262 breast cancer patients undergoing neoadjuvant therapy and treated by axillary lymph node dissection.

Characteristics	ypN0-mi-1 (n = 168)	ypN2-3 (n = 94)	Univariate Analysis *p*-Value	Multivariate Analysis *p*-Value OR (95% CI)
**Demographics**				
Age [years, mean (SD)]	52.3 (11.2)	54.5 (13.1)	0.149	-
Menopausal status				
Pre-menopausal	72 (42.9%)	32 (34.0%)	0.162	-
Post-menopausal	96 (57.1%)	62 (66.0%)	-	
**NAT**				
NAT without anthracycline				
Yes	6 (3.6%)	6 (6.4%)	0.296	-
No	162 (96.4%)	88 (93.6%)	-	
Completed cycles				
Yes	148 (88.1%)	79 (84.0%)	0.355	-
No	20 (11.9%)	15 (16.0%)	-	
Immunotherapy				
Yes	78 (46.4%)	20 (21.3%)	<0.001 ^a^	0.401 1.837 (0.444–7.602)
No	90 (53.6%)	74 (78.7%)	-	-
**Pre-operative staging**				
Single nodule				
Yes	122 (72.6%)	68 (72.3%)	0.961	-
No	46 (27.4%)	104 (27.7%)	-	
Dimension pre-NAT [mm, mean (SD)]	34.3 (17.3)	39.9 (18.2)	0.034 ^a^	0.655 1.006 (0.979–1.034)
Stage pre-NAT				
cT1-2	130 (77.4%)	61 (64.9%)	0.029 ^a^	0.507 1.466 (0.473–4.540)
cT3-4	38 (22.6%)	33 (35.1%)	-	-
cN0	18 (10.7%)	3 (3.2%)	0.031 ^a^	0.013 ^a^ 7.697 (1.537–38.550)
cN+	150 (89.3%)	91 (96.8%)	-	-
**Tumor**				
Subtype				
HR+/HER2−	55 (32.7%)	65 (69.1%)	<0.001 ^a^	0.003 ^a^ 3.945 (1.602–9.716)
Non-HR+/HER2−	113 (67.3%)	29 (30.9%)	-	-
HER2+	72 (42.9%)	12 (12.8%)	<0.001 ^a^	0.183 0.343 (0.071–1.656)
Non-HER2+	96 (57.1%)	82 (87.2%)	-	-
Triple-negative	41 (24.4%)	17 (18.1%)	0.237	-
Non-triple-negative	127 (75.6%)	77 (81.9%)	-	
Histotype				
Ductal	157 (93.5%)	82 (87.2%)	0.088	-
Lobular	11 (6.5%)	12 (12.8%)	-	
Dimension post-NAT [mm, mean (SD)]	12.9 (17.8)	29.9 (32.7)	<0.001 ^a^	<0.001 ^a^ 1.043 (1.021–1.066)
**Surgery**				
BCS	73 (43.5%)	38 (40.4%)	0.634	-
Mastectomy	95 (56.5%)	56 (59.6%)	-	
SLNB followed by ALND	59 (35.1%)	28 (29.8%)	0.379	-
Direct ALND	109 (64.9%)	66 (70.2%)	-	
Number of evaluated non-SLNs				
≤12	71 (42.3%)	28 (29.8%)	0.062	-
>12	97 (57.7%)	66 (70.2%)	-	

Footnotes: OR: Odds ratio, 95%CI: 95% Confidence interval, SD: Standard deviation, NAT: Neoadjuvant therapy, HR: Hormone receptor, HER2: HER2 evaluated either on immunohistochemistry or on in situ hybridization, according to the ASCO CAP guidelines, BCS: Breast-conserving surgery, SLNB: Sentinel lymph node biopsy, ALND: Axillary lymph node dissection, SLN: Sentinel lymph node, ^a^: Statistically significant.

## Data Availability

Data supporting reported results can be found in Appendix A.

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
