# Peer review of "Predictors of High-Burden Residual Axillary Disease After Neoadjuvant Therapy in Breast Cancer"

_cancers, 2025, doi:10.3390/cancers17101596_

Round 1
Reviewer 1 Report
Comments and Suggestions for Authors
Please consider the following major and minor comments before resubmission.
- The introduction section is very concise. Need to add the need of research.
- All the references in text need to be in correct format.
- Statistical analysis results need to be shown in the result section clearly.
- The discussion section looks very bulky. Talk on specific points, not everything together.
Author Response
The introduction section is very concise. Need to add the need of research.
Reply: We thank the Reviewer for their thoughtful suggestion.
In response, we have revised the Introduction section to clearly articulate the need for this research, highlighting the current clinical challenge in identifying patients at risk for high residual axillary disease after NAT.
All the references in text need to be in correct format.
Reply: We thank the Reviewer for this observation.
We have carefully reviewed the entire manuscript and confirmed that all in-text citations are correctly formatted in accordance with the journal’s guidelines, using numerical references in square brackets (e.g., [1], [2]).
Statistical analysis results need to be shown in the result section clearly.
Reply: We thank the Reviewer for the comment.
We respectfully point out that statistical analysis details are clearly reported in the dedicated “Statistical Analysis” subsection of the Materials and Methods section.
We hope this addresses the concern appropriately, and we remain available for further clarifications if needed.
The discussion section looks very bulky. Talk on specific points, not everything together.
Reply: We appreciate the reviewer’s feedback on the structure of the Discussion section.
In response, we have divided the section into clearly labeled thematic paragraphs, each addressing specific aspects of the findings and their comparison with existing literature. We believe this restructuring improves clarity and readability while maintaining the depth of analysis. Thank you for this helpful suggestion.
Reviewer 2 Report
Comments and Suggestions for Authors
The manuscript "Predictors of High-Burden Residual Axillary Disease After Neoadjuvant Therapy in Breast Cancer" addresses a highly important issue for practical oncological interventions, written in good language, technically close to perfection. I recommend to accept it without revisions, a few grammar bugs (like "was-were") can be easily fixed by copy-editors. The only serious limitation of the study (relatively small cohort sizes due to to the data collected from only one location) has been fully declared in the Discussion. Therefore, there is no doubt that this paper will be a useful contribition to the field. The only advisory remark that I would like to give to the authors: perhaps in any future paper(s) it should be worth trying to biuld predictory models separately for major breast cancer types (HER2+, TNBC, etc ) so that to yield higher AUCs.
Author Response
We sincerely thank the Reviewer for the positive and encouraging comments on our manuscript, as well as for recognizing the relevance of our study to current oncological practice.
We are also grateful for the valuable suggestion regarding the development of subtype-specific predictive models in future studies. We fully agree that such analyses could enhance predictive accuracy and provide more tailored clinical insights for each breast cancer subtype. This will certainly be a key consideration in our future research endeavors.
Thank you again for your thoughtful and constructive feedback.
Reviewer 3 Report
Comments and Suggestions for Authors
This is an interesting paper on a challenging topic in breast cancer and, although it does not add much to current knowledge, it reinforces much of the data from other series.
There are two aspects of the series the authors should clarify or explain in my opinion:
The data were obtained from a "prospectively maintained institutional database" between 2006 and 2023, and only 262 cases of cN+ treated with NAT were recorded. This represents an average of 15 cases per year (less than 2 per month). Was a larger sample selected? If so, what selection criteria were used?
After reviewing it, I wonder why more than 62.2% of patients were treated with hormone therapy if only 45.8% were HR+HER-.
Author Response
This is an interesting paper on a challenging topic in breast cancer and, although it does not add much to current knowledge, it reinforces much of the data from other series.
There are two aspects of the series the authors should clarify or explain in my opinion:
The data were obtained from a "prospectively maintained institutional database" between 2006 and 2023, and only 262 cases of cN+ treated with NAT were recorded. This represents an average of 15 cases per year (less than 2 per month). Was a larger sample selected? If so, what selection criteria were used?
Reply: We thank the Reviewer for this important observation regarding the number of cases included in our analysis.
We would like to clarify that this retrospective study specifically analyzed all consecutive breast cancer (BC) patients who underwent neoadjuvant therapy (NAT) followed by axillary lymph node dissection (ALND) at the Breast Unit of IRCCS Humanitas Research Hospital, Milan, Italy, between October 2006 and December 2023. The inclusion criterion of undergoing ALND was central to our study design, as our primary objective was to identify predictors of high-burden residual axillary disease (ypN2-3), a condition that can only be reliably assessed in patients who received a full ALND.
It is important to note that not all patients treated with NAT proceeded to ALND. In fact, the majority of BC patients treated with NAT at our Breast Unict actually achieved axillary pathologic complete response (ypN0) and were therefore spared ALND. As a result, the size of our cohort may appear limited in absolute terms but represents a highly selected and clinically relevant population.
We hope this clarifies the rationale behind the cohort size and selection criteria.
After reviewing it, I wonder why more than 62.2% of patients were treated with hormone therapy if only 45.8% were HR+HER-.
Reply: We thank the Reviewer for this insightful observation.
The apparent discrepancy between the 62.2% of patients receiving adjuvant endocrine therapy and the 45.8% with HR+/HER2− tumors is explained by the presence of 43 triple-positive patients (HR+/HER2+). These patients are typically treated with both anti-HER2 targeted therapies and endocrine therapy in the adjuvant setting.
We hope this clarifies the rationale for the endocrine therapy prescription rates.